# Efficiency Comparative Approach of Plant-Produced Monoclonal Antibodies against Rabies Virus Infection

**DOI:** 10.3390/vaccines11081377

**Published:** 2023-08-17

**Authors:** Boonlert Lumlertdacha, Bancha Mahong, Kaewta Rattanapisit, Christine Joy I. Bulaon, Thiravat Hemachudha, Waranyoo Phoolcharoen

**Affiliations:** 1Department of Animal Diagnosis and Investigation, Queen Saovabha Memorial Institute, Thai Red Cross Society, Bangkok 10330, Thailand; qsmibld@yahoo.com; 2Department of Biological Products, The Government Pharmaceutical Organization, 75/1 Rama VI Rd., Ratchet, Bangkok 10400, Thailand; bancha.ma@gpo.or.th; 3Center of Excellence for Plant-Produced Pharmaceuticals, Pharmacognosy and Pharmaceutical Botany Department, Faculty of Pharmaceutical Sciences, Chulalongkorn University, Bangkok 10330, Thailand; 4Faculty of Medicine, Chulalongkorn University, Bangkok 10330, Thailand; fmedthm@gmail.com

**Keywords:** rabies virus, post-exposure prophylaxis, monoclonal antibody, *Nicotiana benthamiana*, plant produced antibody cocktail

## Abstract

Rabies encephalitis is a fatal zoonotic viral disease caused by the neurotropic rabies virus. It remains a major public health concern as it causes almost 100% fatality and has no effective medication after the onset of the disease. However, this illness is preventable with the timely administration of effective post-exposure prophylaxis (PEP) consisting of the rabies vaccine and passive immune globulins (HRIG and ERIG). Recently, conventional PEP has been shown to have many limitations, resulting in little support for these expensive and heterologous globulins. Monoclonal antibody (mAb) production via recombinant technology in animal and human cell cultures, as well as a plant-based platform, was introduced to overcome the costly and high-tech constraints of former preparations. We used transient expression technology to produce two mAbs against the rabies virus in *Nicotiana benthamiana* and compared their viral neutralizing activity in vitro and in vivo. The expression levels of selective mAbs E559 and 62-71-3 in plants were estimated to be 17.3 mg/kg and 28.6 mg/kg in fresh weight, respectively. The plant-produced mAbs effectively neutralized the challenge virus CVS-11 strain in a cell-based RFFIT. In addition, the combination of these two mAbs in a cocktail protected hamsters from rabies virus infection more effectively than standard HRIG and ERIG. This study suggests that the plant-produced rabies antibody cocktail has promising potential as an alternative biological to polyclonal RIG in rabies PEP.

## 1. Introduction

The rabies virus (RABV) causes encephalomyelitis in humans and animals. One of the major challenges of RABV infection, such as undetectable levels of virus-neutralizing antibodies in infected individuals, has a significant impact on this disease’s death toll [1,2,3]. However, the prompt administration of post-exposure prophylaxis (PEP) after certain recognized exposures has proven to be highly effective in preventing the disease [4,5]. The recommended PEP treatment involves immediate wound cleansing, active immunization with multiple doses of the rabies vaccine, and passive immunization with rabies immune globulin (RIG) derived from hyperimmunized humans (HRIG) or horses (ERIG). The goal of PEP is to prevent RABV from entering the nervous system, which would eventually lead to death [6]. Despite advances in the development of rabies biologicals, both HRIG and ERIG are expensive and in limited supply, restricting access to RIG for sufficient PEP. Furthermore, RIG has variable efficacy across batches, is susceptible to contamination with blood-borne adventitious agents, and, in the case of ERIG, can cause some adverse allergic responses [7]. For these reasons, several works of research have set out to seek different alternatives.

The World Health Organization Rabies Collaborating Centers (WHO RCCs) have identified five mouse-derived monoclonal antibodies (mAbs) with RABV neutralizing activity [8], four of which target the antigenic site II of the viral glycoprotein 1, one of which targets antigenic site I [9]. They are being developed as potential components of safer next-generation PEP. Rabies-neutralizing mAbs are appealing alternatives to standard RIGs due to their high specificity and potency, as well as their clinical and commercial success [8,10]. Previous research has shown that rabies-specific mAbs can protect rodents after RABV infection [11,12,13]. Among the therapeutic mAbs identified, E559 had the broadest RABV neutralization breadth and greatest potency [8,14]. Meanwhile, other mAbs include M727-5-1, M777-16-3, 1112-1, and 62-71-3.

MAb-based therapies have a number of advantages, such as improved consistency and safety, and, with humanization, better patient tolerance [15,16]. Plants are a viable and competent platform for producing mAbs due to their low cost of production, capability to assemble multimeric proteins, and high scalability [17,18]. Prior studies have demonstrated that the transient expression of rabies mAbs (R-mAb) E559 and 62-71-3 in tobacco (*Nicotiana bethamiana*) leaves [9,14,19,20,21], and plant-produced R-mAbs efficiently neutralized RABV in vitro and conferred protection in a hamster challenge model [22]. More importantly, these plant-produced R-mAbs had higher protective activity than HRIG in vivo. However, since individual neutralizing R-mAbs are specific to different epitopes on the RABV glycoprotein, a cocktail of two or more non-competing R-mAbs targeting non-overlapping epitopes could provide greater specificity and potency [22]. Thus far, the WHO has proposed cocktails consisting of mAb 62-71-3, which binds to an epitope in antigenic site I, and one of the mAbs E559, M727-5-1, M777-16-3, or 1112-1, binds to an epitope in antigenic site II. Given the broad neutralization range of E559, its inclusion in a RIG-replacement cocktail is being considered [14]. The use of E559 and 62-71-3 in a cocktail of neutralizing R-mAbs, as each mAb targets different epitopes of the RABV glycoprotein, could provide coverage against a wide spectrum of RABV isolates, prevent viral escape, and have a comparable or better potency than commercial RIG. Previous research has presented a basis for the development of an R-mAb cocktail (E559 and mAb 62-71-3) [22]; however, it has yet to be proven whether multi-mAb treatment can increase rabies viral neutralization in an animal model.

The current study describes plant-based recombinant mAb production and the functional characterization of R-mAbs, E559 and 62-71-3, in vitro and in vivo. These two R-mAb candidates were expressed in *Nicotiana benthamiana* and purified using protein A’s affinity chromatography. Plant-produced R-mAbs were tested for in vitro efficacy and showed the effective viral neutralization of the standard challenge virus (strain CVS11) in a cell-based RFFIT assay. In addition, plant-expressed R-mAbs demonstrated in vivo efficacy in a hamster post-exposure viral challenge using Thailand-isolated rabies virus and isolate 338 PJm [23], with E559 showing high potency and 62-71-3 mAb being at least as effective as standard HRIG. Moreover, the R-mAb cocktail had enhanced in vivo potency compared to individual plant-based R-mAb and conventional RIG. Reduction determinants of clinical rabies are essential parameters for biological efficacy.

## 2. Materials and Methods

### 2.1. Plant-Produced Rabies Monoclonal Antibodies in Nicotiana benthmiana

*Agrobacterium tumefaciens* cells containing plasmids of E559 [14] and 62-71-3 [19] genes were obtained from the Research Centre for Infection and Immunity, Division of Clinical Sciences (St George’s University of London, London, UK). About 4–5-week-old *Nicotiana benthamiana* plants were grown at 16 h light/8 h dark cycle and were used for the experiment. Bacterial cultures were cultivated in an antibiotic-selective Luria-Bertani broth supplemented with 25 mg/L rifampicin, 50 mg/L carbenicillin, and 50 mg/L kanamycin at 28 °C with a continuous shake (200 rpm) until the OD_600_ was approximately 0.1 and were subsequently used for agro-infiltration into *N. benthamiana* using the vacuum chamber. The infiltrated plants were incubated at 28 °C for 7 days, and only the infiltrated leaves were harvested. The plant produced E559 and 62-71-3 mAbs, which were extracted by blending the leaves with Phosphate Buffered Saline (PBS). The crude extract solution was then centrifuged and clarified, and the supernatant was purified further using protein A’s affinity chromatography column. Briefly, the crude lysate was loaded onto a column packed with protein A beads (Expedeon, Cambridge, UK). The proteins were washed, and the recombinant E559 and 62-71-3 mAbs were eluted using an elution buffer at pH 2.7 and were immediately neutralized to the final pH 7.4. The purified proteins were concentrated by ultrafiltration using an Amicon^®^ Ultra Centrifugal Filter (Merck, Darmstadt, Germany). The molecular weights of plant-produced E559 and 62-71-3 were estimated by SDS-PAGE and the Western blot technique, and specific antibodies, namely goat anti-human IgG-HRP and goat anti-human kappa-HRP, were used for detection.

### 2.2. Analytical Rabies Monoclonal Antibodies Concentration

The protein concentrations of plant-produced E559 and 62-71-3 were determined using a sandwich ELISA method [24]. Briefly, a 96-well plate was coated with goat polyclonal Ab to Human IgG (Abcam, Cambridge, UK) as the capture antibody overnight at 4 °C. The plate was washed three times with PBS containing 0.05% Tween 20 (PBS-T) and blocked with 5% skim milk. The commercial human IgG1 (Abcam, Cambridge, UK) was used as the standard and was serially diluted two-fold at various concentrations up to 125.000 ng/mL, while E559 and 62-71-3 antibodies were also diluted to optimize absorbance and fill within the standard absorbance range. Every sample concentration was prepared in duplicate and added to the plate. After incubation, the plate was washed with PBS-T to decrease the non-specific binding of the antigen or antibody. The goat anti-human Kappa-HRP antibody (Southern Biotech, Birmingham, AL, USA) was used for detection and added to each well. After being incubated with the detection antibody, the plate was washed with PBS-T three times. The 3,3′,5,5′-Tetramethylbenzidine (TMB) solution was prewarmed to room temperature, added to the plate, and incubated until the color of the solution changed. The enzymatic reaction was stopped with 1 M H_2_SO_4_ and color development was measured at OD_450_ using a microplate reader. The absorbances of standard IgG1 were exploited to construct the standard curve, which was then used to calculate the concentrations of E559 and 62-71-3, respectively.

### 2.3. Rapid Fluorescent Focus Inhibition Test (RFFIT)

We measured the antigenic unit of monoclonal antibodies using the RFFIT technique, as previously described [25,26]. A five-fold serial dilution of plant-expressed mAb was prepared in MEM-10 (Eagle’s minimum essential medium with 10% fetal bovine serum), and 50 µL of each test-mAb dilution was analyzed in an 8-well tissue culture chamber slide (LabTek^®^, Thermo Fisher Scientific, Waltham, MA, USA). A constant amount of 100 µL of the standard challenge virus (CVS-11) containing 1 × 10^4^ FFD50 (50% Focus Forming Dose) was added to each well, and the mixture was allowed to react for 90 min in a CO_2_ incubator at 37 °C. Mouse neuroblastoma (MNA) cells were obtained from the Department of Medical Sciences, Ministry of Public Health, Thailand, and approximately 5 × 10^4^ MNA cells were added to each well. The slides were incubated for another 20 h in a CO_2_-controlled incubator at 37 °C and were then fixed with ice-cold 80% acetone and stained with FITC-conjugated anti-RABV antibodies (Fujirebio Diagnostics, Inc., Malvern, PA, USA) for 30 min at 37 °C. Twenty microscopic fields in each well were observed under a fluorescent microscope with a 40× objective, and the presence of fluorescing cells was counted. The antigenicity of mAb was calculated according to the Reed–Muench formula [27]. The values were compared to those of the reference serum (obtained from the National Institute for Biological Standards and Control, Herts, UK) and normalized to international units (IU/mL). The lower limit of detection was 0.1 IU/mL.

### 2.4. Ethics Statement

The project, entitled the “Efficacy study of tobacco produced monoclonal antibody against rabies virus infection in vivo” with the ACU number QSMI-ACUC-10-2020, was approved on 1 March 2021 by the Queen Saovabha Memorial Institute Animal Care and Use Committee (IACUC), Thai Red Cross Society.

### 2.5. Rabies Virus (RABV) Preparation

A canine RABV, TDRBV-QSMI2 (isolate 338 PJm), was originally isolated from a naturally infected dog at our Rabies Diagnostic Laboratory Facilities. The virus suspension was prepared from the homogenate and centrifuged 20% (*w*/*v*) of the dog brain to remove debris. The supernatant was collected and stored at −80 °C. About 30 µL of the dog-brain suspension was inoculated intracranially into weaning mice. After exhibiting typical symptoms of rabies, the mice were euthanized, and their brains were collected. Another seven serial passages were performed via the intracranial inoculation of weaning mice to characterize the onset of clinical signs.

### 2.6. Hamsters Challenge

Each experimental group included ten 6-week-old Golden Syrian hamsters (*Mesocricetus auratus*) with body weights ranging from 190 to 210 g. The hamsters were anesthetized with isoflurane in an induction chamber and, afterwards, infected by an IM route with 43 × MICLD_50_ of TDRBV-QSMI2 via direct inoculation into the right gastrocnemius muscle.

After 4 h of post-infection, hamsters were given a rapid anesthetic inhaled isoflurane in the induction chamber to administer the parenterally PEP. Animals were vaccinated with Verorab^®^ (inactivated rabies vaccine; batch T1A621M manufactured by Sanofi Pasteur, Lyon, France) via IM administration on the left gastrocnemius on days 0, 3, 7, and 14 with an allometric conversion dose to 20 µL. The allocated dosage of HRIG (batch HRG-63001 containing 5 mL vial manufactured by National Blood Centre, Thai Red Cross Society, Thailand, comprising 172.4 IU/mL) was 20 IU/kg, and ERIG (batch RF-00319 containing 200 IU/mL manufactured by the Queen Saovabha Memorial Institute, Thai Red Cross Society, Bangkok, Thailand) was 40 IU/kg. Experimental hamsters were observed twice daily. The humane endpoint of the study was defined as the appearance of hind limb paralysis on one or both sides, and the experimental endpoint was determined based on the clinical signs observed. Brain samples were tested using the direct fluorescent antibody technique (DFA) to determine the presence of the virus. Hamsters that survived the infection for 28 days were euthanized with carbon dioxide.

### 2.7. Allometric Conversion Dose

The dosages for the PEP protocol were approved for human use and should not be extrapolated to animals directly without proper modification. A previous study [28] suggested that converting the calculation between the human equivalent dose (HED) and species varied the animal equivalent dose (AED) by allocating all the dosages of biologicals with body surface area (BSA) normalization.

### 2.8. Formula for Dose Transferation Based on BSA

Animal equivalent dose (AED) = Human equivalent dose (HED) × Km value

Km value = constant value unique to each species calculated for a range of body weight
using Km = 100/K × W0.33

(Km for adult human = 37, Km for hamster = 5).

### 2.9. Statistical Analyses

Data were analyzed in a simple survival curve using the Kaplan–Meier method and were compared statistically using the log-rank test. GraphPad Prism was used to plot the graphs, and a *p*-value < 0.05 was considered statistically significant.

## 3. Results

### 3.1. Expression and Purification of Plant-Produced R-mAbs E559 and 62-71-3

R-mAbs E559 and 62-71-3 were transiently produced in *Nicotiana benthamiana* and purified by protein A’s affinity chromatography column with yields of 17.3 and 28.6 mg/kg of fresh weight after purification. Accordingly, the purity and size of purified plant-produced R-mAbs were determined by SDS-PAGE and Western blot. Under reducing conditions, the size of the R-mAb E559 heavy chain protein correlated to the theoretical size with approximately 50 kDa, but the light chain size displayed two protein bands that were slightly larger than 25 kDa (Figure 1b, lane 1). On the other hand, purified plant-produced E559 displayed a protein band around 150 kDa under the non-reducing condition (Figure 1f, lane 3). In Western blot analysis, the heavy chain and light chain of purified plant-produced E559, as well as its assembly into the whole antibody with two heavy and light chains, was confirmed by anti-Kappa (Figure 1c,g, lanes 1 and 3) and anti-Gamma (Figure 1d,h, lanes 1 and 3) under reducing and non-reducing conditions. These results are similar to the human IgG control standard (Figure 1a,e, lane A). Purified plant-produced R-mAb 62-71-3 was also characterized using the same methods under both reducing (Figure 1b, lane 2) and non-reducing (Figure 1f, lane 4) conditions. The protein size and pattern with Instant Blue^®^ staining are comparable to standard human IgG, and Western blot results showed that both the anti-Kappa and anti-Gamma antibody could detect plant-produced 62-71-3 mAb at expected sizes under reducing (Figure 1c,d, lane 2) and non-reducing (Figure 1g,h, lane 4) conditions. Based on Instant Blue staining data (Figure 1b,f, lanes 1–4), plant-derived R-mAbs E559 and 62-71-3 were successfully purified (major bands indicated by black arrows) with approximately more than 80% purity based on visual inspection.

### 3.2. In Vitro Study of Plant-Produced R-mAbs E559 and 62-71-3 on RFFIT

The two plant-produced R-mAbs, E559 and 62-71-3, are known to target different complementary sites on the RABV glycoprotein [8,9,29]. In the present study, it was essential to determine whether the tobacco-based R-mAbs were functional and incongruent with similar studies [22], as well as to estimate the antibody concentration against RABV in terms of international units/mL (IU/mL).

To test and compare the coverage of plant-produced mAbs to that of HRIG on a laboratory isolate of RABV (strain CVS11), a rapid fluorescent focus inhibition test (RFFIT) method was performed. The results revealed antibody concentrations of 807.6 IU/mL for plant-produced R-mAb E559 and of 546.2 IU/mL concentration for plant-produced R-mAb 62-71-3, respectively. Meanwhile, conventional HRIG had 1 IU/mL antibody concentration. IU/mL data were used to calculate the antibody ratio needed to make an antibody cocktail for use in animal administration. Consequently, we varied the ratios of two plant-produced R-mAbs (E559:62-71-3) to neutralize RABV as 1:1, 1:2, and 2:1 and found that the endpoints of the different cocktail ratios had no significance. As a result, we formulated the cocktail E559:62-71-3 as 1:1, and the volume was adjusted to 50 µL per animal, with either 20 IU/kg or 40 IU/kg where applicable.

### 3.3. In Vivo Study of Plant-Produced R-mAbs E559 and 62-71-3

To assess the potency of plant-produced R-mAbs in vivo, ten hamsters in each group were infected via the IM route with RABV isolate 338 PJm at a dose of 43 *×* MICLD_50_ by direct inoculation into the right gastrocnemius muscle for 4 h prior to the postexposure prophylaxis (PEP). Animals were vaccinated with Verorab^®^ via the IM route on the left gastrocnemius with an allometric conversion dose of 20 µL. The allocated dosage was 20 IU/kg, and ERIG was 40 IU/kg. In this assay, all hamsters were infected with the challenge RABV and then were given either a mock PEP, vaccine, or R-mAbs. Based on these results, the survival rate after the following treatments was 0% (Table 1). Hamsters treated with either PBS (group 1) or vaccine only (group 2) died after 13 days. Meanwhile, animals that received cocktail R-mAbs at 20 IU/kg (group 3) and at 40 IU/kg (group 4) died after 19 and 14 days, respectively. In the case of the combination regimen, treatment with both the vaccine and passive globulins resulted in higher survival rates, with 50% of hamsters surviving after receiving conventional PEP with HRIG (group 5) and 30% surviving after receiving PEP with ERIG (group 6). Notably, the combined treatment of the vaccine with the plant-produced R-mAb cocktail (E559:62-71-3) also increased the survival rates of challenged animals by up to 90%. Survival rates against the RABV challenge after vaccine treatment with cocktail R-mAbs at 20 IU/kg (group 7) or at 40 IU/kg (group 8) were 60% and 90%, respectively. Moreover, hamsters treated with vaccines in combination with either E559 mAb (group 9) or 62-71-2 mAb (group 10) had survival rates of 70% and 50% (Table 1).

The in vivo potency of either the vaccine or plant-produced R-mAb (E559:62-71-3) cocktail alone was tested in a challenge experiment with hamsters infected with the RABV isolate 338 PJm (Figure 2). Based on these results, 50% of animals treated solely with vaccines did not survive beyond 8 days post-infection (dpi). Later on, all animals in both the infected control group and the vaccine-treated group died at 13 dpi. Meanwhile, animals given either 20 or 40 IU/kg of the plant-produced R-mAb cocktail showed 50% protection at 12 dpi, which was slightly greater than those seen with the control and vaccine groups at the exact time point. Nonetheless, 100% of animals administered with 40 IU/kg of the plant-produced R-mAb cocktail died within 14 dpi, whereas 100% of animals given 20 IU/kg of the plant-produced R-mAb cocktail died after 19 days. Collectively, these findings show that either the vaccine or antibody treatment on its own is ineffective at neutralizing the RABV isolates 338 PJm in vivo.

Here, we also present a head-to-head comparison of a plant-produced R-mAb cocktail with conventional PEP. In combination with the vaccine, antibody cocktails at 40 and 20 IU/kg showed higher survival rates than the combination of the vaccine plus HRIG or ERIG (Figure 3). The R-mAb cocktail improved the efficacy of the rabies vaccine, with approximately 90–100% of challenged hamsters still alive after 8 days of the RABV infection compared to only 70–80% of animals that received HRIG or ERIG. At the study endpoint, the survival rate from the R-mAb cocktail at 40 IU/kg remained at 90%, whereas the survival rates from HRIG or ERIG dropped to 50% and 30%, respectively.

However, there was no statistically significant difference between the survival curves of the group receiving the vaccine and R-mAb cocktail at 40 IU/kg and the group receiving the vaccine and R-mAb cocktail at 20 IU/kg (*p* = 0.10).

The efficiency of the plant-produced mAbs and R-mAb cocktail was also compared, with the antibody cocktail outperforming individual mAbs in terms of protection (Figure 4). In brief, among all plant-produced mAb treatments, the R-mAb cocktail at 40 IU/kg in combination with the rabies vaccine demonstrated the best protective effect with 90% animal survival. The survival curve in the group receiving the vaccine and R-mAb cocktail at 40 IU/kg differed from those of the group receiving the vaccine and HRIG (*p* = 0.03) and vaccine and ERIG (*p* = 0.004). Surprisingly, the vaccine plus E559 mAb showed higher survival rates than 62-71-3 mAb and even the R-mAb cocktail, all at the same dose of 20 IU/kg. These findings suggest that E559 mAb is potent and increases the vaccines’ efficacy in challenged hamsters. Overall, the results show that the R-mAb cocktail at 40 IU/kg is superior to the conventional RIG and that plant-produced 62-71-3 mAb at 20 IU/kg is at least as effective as the standard HRIG and vaccine treatment.

## 4. Discussion

The WHO recommends the use of at least two mAbs in addition to the rabies vaccine for rabies post-exposure prophylaxis (PEP). This conventional PEP prevents viral avoidance, enhances viral neutralization potency, and covers a broad range of viral strains. Currently, only two commercial PEP products, containing an anti-rabies virus (anti-RABV) mAbs are available, and two mAb cocktails are being tested in clinical trials, which are described briefly below. First, the Serum Institute of India Pvt. Ltd. (Pune, India) and MassBiologics (Mattapan, MA, USA) released a mAb called ”Rabishield” which was developed from transgenic mice harboring human immunoglobulin. Then, B-cells were isolated and selected for the highest RABV-specific antibody. However, Rabishield contains only one mAb, which does not meet the WHO recommendations [30]. Second, the Crucell company in the Netherlands developed a human mAb cocktail comprising two mAbs, CR57 and CR4098, derived from transgenic mouse harboring human immunoglobulin. This product has already undergone phase 1 and 2 clinical trials in India, the Philippines, and the USA [31], yet it did not proceed to commercialization. Third, Zydus Cadila developed Twinrab, the first commercial rabies-specific mAb cocktail, which was released in India in 2020. The mAb cocktail contains mAb M777-16-3 and mAb 62-71-3, and the recommended dose is 40 IU/kg [32]. Lastly, Synermore Biologics developed SYN023, a mAb cocktail containing two humanized mAbs: CTB011 and CTB012. The mAb cocktail has the ability to bind and neutralize 20 viruses from a pool of ten China rabies virus isolates and ten USA rabies virus isolates. To date, phase 3 clinical trials for SYN023 have already started.

All anti-rabies mAbs, whether commercially available or nearly so, are produced in cell-based systems and are quite expensive, whereas mAbs derived from plant-based systems are manufactured at a low cost. To consider the course of mAb production between the transient protein expression in tobacco leaves and mammalian cells (i.e., CHO cells), the iBio CMO, LLC company, which is a plant-produced vaccine and antibodies company, used a simulation model in the production of crude mAb 1 g/L (mammalian cell) and 1 g/kg (fresh weight tobacco) [33]. Mammalian cells can produce 250 kg/year with 70% of the protein recovery after complete purification, and the course of production is roughly $65 million USD/year for the manufacturing of $250 USD/g. Meanwhile, tobacco can produce 300 kg/year with 65% protein recovery, and the course of production is $33 million USD/year for the manufacturing of $131 USD/g. These data suggest that in an equal amount of plant-produced and mammalian cell-produced mAb production, plant-produced is around 50% cheaper. Thus, plant-produced R-mAb, for instance, from tobacco leaves, is another interesting technology due to the above-mentioned high protein recovery and low production cost.

The production and development of mAbs against RABV have continuously progressed. Despite the fact that no products have been approved, plant-produced R-mAbs 62-71-3 and E559 from *Nicotiana benthamiana* leaves with concentrations of 546 IU/mL and 807 IU/mL, respectively, are promising for further development. Since these two antibodies have different binding sites, they can prospectively bind to RVG at the same time. The combination of these R-mAbs could improve the neutralization of a broader spectrum of RABV and minimize the likelihood of viral escape [29]. The WHO recommends the use of HRIG at a dose of 20 IU/kg and ERIG at a dose of 40 IU/kg for patients. Thus, both of our plant-produced R-mAbs could be potential and interesting candidates to further develop for patient use. Previous studies have [9,12] demonstrated that tobacco-produced E559 and 62-71-3 are more effective in vivo than HRIG, which is consistent with our findings showing the mAb E559 dosing at 20 IU/kg is more potent than HRIG and that of mAb 62-71-3 dosing at 20 IU/kg, which had a comparable neutralizing potency to HRIG. Nonetheless, the combination of plant-produced R-mAbs (such as E559 and 62-71-3) for their neutralizing effect against the rabies virus has not been tested. Thus, our study is the first to show that cocktail tobacco-produced E559 and 62-71-3 increased animal survival rates more than either of them.

## 5. Conclusions

We have demonstrated the feasibility of the rapid transient expression and production of the plant-produced E559 and 62-71-3 rabies monoclonal antibody (R-mAbs) in *N. benthamiana*. A cocktail of these R-mAbs was tested for in vitro efficacy in a hamster model challenge and was found to be superior to conventional RIG for postexposure prophylaxis. Altogether, this proof-of-concept study reveals the promising advantage of using a plant expression system to produce specific anti-rabies virus globulin for safer and more effective biologicals in humans.

## Figures and Tables

**Figure 1 vaccines-11-01377-f001:**
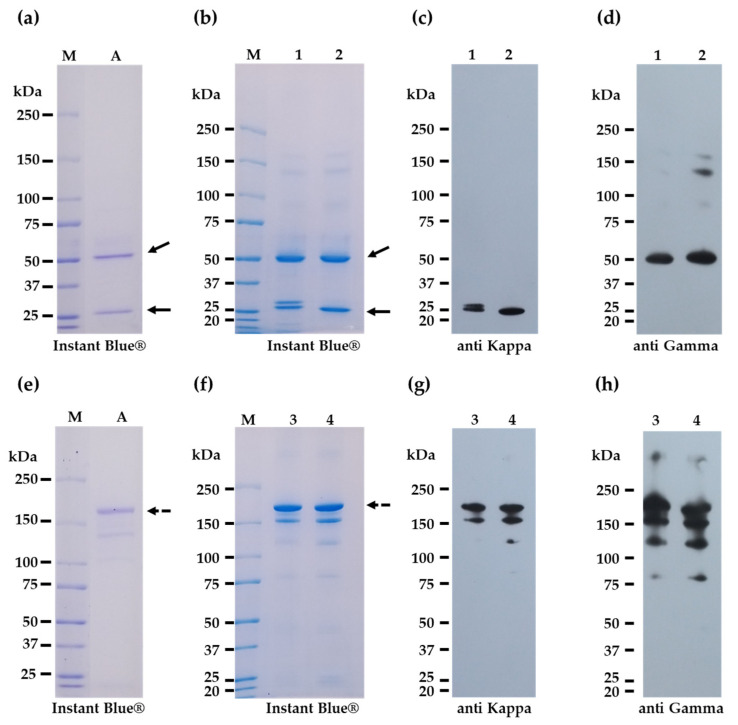
Determination R-mAb size with SDS-PAGE (**a**,**b**,**e**,**f**), and Western blot (**c**,**d**,**g**,**h**). M, protein Marker; A, standard human IgG; 1, E559 reducing condition; 2, 62-71-3 reducing condition; 3, E559 Non-reducing condition; 4, 62-71-3 Non-reducing condition; horizontal bold arrow, heavy chain size; oblique bold arrow, light chain size; dash horizontal arrow; native form R-mAb.

**Figure 2 vaccines-11-01377-f002:**
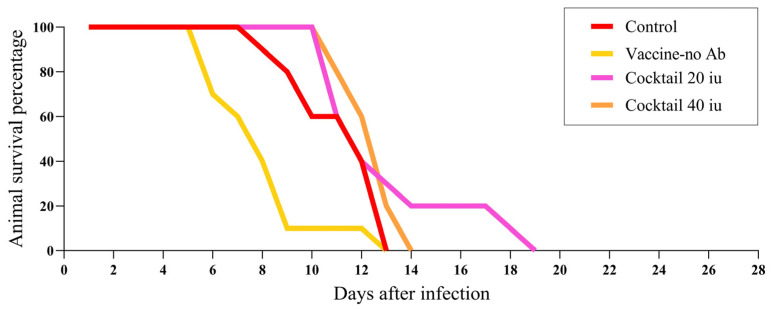
Number of animals survived in groups of the control(mock), vaccine, and the cocktailed plant mAbs.

**Figure 3 vaccines-11-01377-f003:**
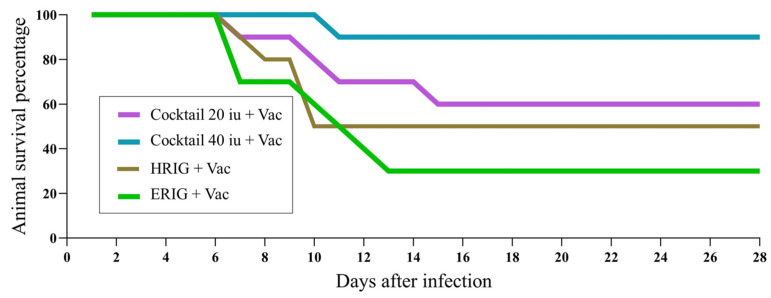
Number of animals that survived in groups of conventional PEP and the cocktailed plant mAbs 20 IU/kg and 40 IU/kg.

**Figure 4 vaccines-11-01377-f004:**
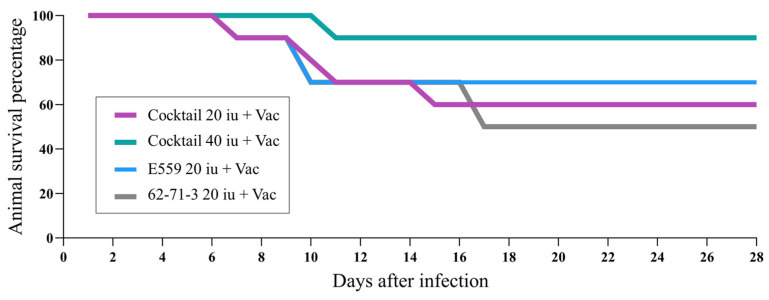
Number of animals that survived in groups of plant produced mAb and the cocktailed plant produced mAbs with vaccination course.

**Table 1 vaccines-11-01377-t001:** Hamster survival rate after the rabies virus isolate 338 PJm infection then administration with R-mAbs.

Group	Vaccination	Antibody * IU/kg	Survival Rate (%)
1	-	-	0
2	4 doses	-	0
3	-	cocktail R-mAb 20 IU/kg	0
4	-	cocktail R-mAb 40 IU/kg	0
5	4 doses	bHRIG 20 IU/kg	50
6	4 doses	cERIG 40 IU/kg	30
7	4 doses	cocktail R-mAb 20 IU/kg	60
8	4 doses	cocktail R-mAb 40 IU/kg	90
9	4 doses	E559 20 IU/kg	70
10	4 doses	62-71-3 20 IU/kg	50

***** All antibodies, including mAb and the cocktail mAbs, were administered one time at day 0 and four hours after RABV infection at the same position of viral infection (right gastrocnemius muscle).

## Data Availability

Not applicable.

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
