# Peer review of "Efficiency Comparative Approach of Plant-Produced Monoclonal Antibodies against Rabies Virus Infection"

_vaccines, 2023, doi:10.3390/vaccines11081377_

Round 1
Reviewer 1 Report
The study of Lumlertdacha et al describes the development and evaluation of plant-produced rabies antibodies. The idea of plantibodies, including rabies plantibodies, is not novel. The antibodies that authors include in their study are well known as well. What brings novelty to this study is a combination of these two technologies together, and evaluation of the result in vitro and in a rodent model. As rabies continues to be a significant public health problem globally, and costs of conventional rabies prophylaxis remain prohibitive in large parts of the globe, an elaboration of the idea to introduce highly efficient rabies plantibodies in human health practice is important, particularly in the developing countries which suffer from rabies most significantly. As such, this preliminary report deserves publication.
I recommend a minor revision and have several suggestions to improve this manuscript.
From y perspective, the animal experiment was not ideal, as no PEP regimen elicited 100% protection of animals from lethal disease. The same was reported previously in such susceptible model as Syrian hamsters, which makes the selection of treatment window very important. This animal model does not recapitulate directly human rabies infection, and as such authors might consider to evaluate shorter windows between 0-2 h post exposure to try and improve PEP efficacy. Alternatively, authors might consider other animal models, e.g. mouse. But this perhaps they would consider for further studies.
All experimental results need statistical evaluation.
Instead of the Protein A purification which is not very selective to immunoglobulins, would authors consider to make constructs of tag-fused immunoglobulin genes, and further specifically purify the fused proteins? To assess purification efficiency, I suggest to implement an additional characterization of the output, e.g. masspec. This obviously would not be suitable for mass production but in small-scale experiments could help to assess better the purity of the product and its components.
Authors need to state whether the RABV isolate they used for challenge, TDRBV-QSMI2, has epitope incompatibility with at least one of the Mabs included in the cocktail. Based on the results, I assume that both epitopes are present on viral glycoprotein but this needs to be described.
Abstract: Language needs a substantial improvement.
Lines 33-47: Language needs an improvement.
Lines 66-68: I think authors mean that in the case if Mab binding epitope is not conserved in certain viruses, these viruses would not be neutralized, and as such, a cocktail of two or more non-competitive Mabs targeting non-overlapping epitopes would warrant a better degree of confidence. If I am wrong, I am in the camp of readers who would equally misunderstand what authors wanted to say here.
Lines 68-69: these Mabs do not have specificity to antigenic sites I or II, their binding epitopes are located within these antigenic sites. Here and elsewhere, authors need to be clear and specific in the terminology they use.
Line 87: What are “reduction determinants of clinical rabies”?
Line 164: What are “virogenic properties”?
Line 173: I do not understand this: “Hamsters were recuperated after 4 h of post infection”.
Lines 174-175: what that “experimentally PEP” was? I see it in lines 247-252, but in the layout that authors chose (M+M first, Results and Discussion next), these technical details must be explained here, and only partially/essentially repeated in Results.
Line 179: did hamsters really “survived the infection for 28 days”?
Lines 190-193: Authors need to check their explanation of this equation. The statement in Line 190 “Animal equivalent dose (AED) = Human equivalent dose (HED) x Km value” cannot be in agreement with the statement in line 193 “Km for adult human = 37, Km for hamster = 5”. Usually, the most simple and conventional way is to divide the animal weight by human weight, and calculate the corresponding drug dose as in these examples: https://www.ncbi.nlm.nih.gov/pmc/articles/PMC4804402/.
Figure 1 and caption (lines 217-222): there are no labels in this figure, impossible to follow the caption. Also, in line 218 it says E550 although everywhere also it is about E559.
Table 1: no need in this table, it can be replaced by one short sentence within the paragraph above.
Lines 144-310: was there any statistical difference, either in survivorship or at least in the duration of incubation or clinical periods between the groups?
Figures 2, 3, 4: MS Excel used for depicting the survival curves is certainly inferior to other software (e.g. Graphpad Prism) in graphing survival curves. In these figures, it is hard to understand the gradual decreases of “Animal survival percentage” along several days observed in some groups. Also, as I have a hard time to imagine a survival of 120%, I do not understand why not to limit the axis to 100%. I recommend to re-do all figures as Kaplan-Meier plots and to compare different Kaplan–Meier curves by the log rank test, and the Cox proportional hazards test to show significant difference between treatment variants.
Lines 312-331: language needs an improvement.
All recommendations about a language improvement made in the body of review.
Author Response
Reviewer 1
The study of Lumlertdacha et al describes the development and evaluation of plant-produced rabies antibodies. The idea of plantibodies, including rabies plantibodies, is not novel. The antibodies that authors include in their study are well known as well. What brings novelty to this study is a combination of these two technologies together, and evaluation of the result in vitro and in a rodent model. As rabies continues to be a significant public health problem globally, and costs of conventional rabies prophylaxis remain prohibitive in large parts of the globe, an elaboration of the idea to introduce highly efficient rabies plantibodies in human health practice is important, particularly in the developing countries which suffer from rabies most significantly. As such, this preliminary report deserves publication.
I recommend a minor revision and have several suggestions to improve this manuscript.
From y perspective, the animal experiment was not ideal, as no PEP regimen elicited 100% protection of animals from lethal disease. The same was reported previously in such susceptible model as Syrian hamsters, which makes the selection of treatment window very important. This animal model does not recapitulate directly human rabies infection, and as such authors might consider to evaluate shorter windows between 0-2 h post exposure to try and improve PEP efficacy. Alternatively, authors might consider other animal models, e.g. mouse. But this perhaps they would consider for further studies.
Author response:
Similar experiments studied on Syrian hamsters, and few other mammal species, considering that they are susceptible for rabies virus infection introduced intra-muscularly. Mouse nor rat is not specimen of choice for experimental rabies infection intramuscularly. We provided a 4-hours post-exposure in our experiment to imitate a longer hour for patients to seek medical cares in actual.
All experimental results need statistical evaluation.
Author response:
We have added statistical analysis into the manuscript per suggestion.
Instead of the Protein A purification which is not very selective to immunoglobulins, would authors consider to make constructs of tag-fused immunoglobulin genes, and further specifically purify the fused proteins? To assess purification efficiency, I suggest to implement an additional characterization of the output, e.g. masspec. This obviously would not be suitable for mass production but in small-scale experiments could help to assess better the purity of the product and its components.
Author response: Alternatively, tagged affinity chromatography can be used by producing recombinant antibody as a fusion protein containing a terminal affinity tag (e.g., poly-His tag) capable of binding to functionalized resin (e.g., nickel compounds). Regardless, this method results in an antibody with an attached tag, which may affect or interfere with the antibody’s binding affinity for its target antigen. As a result, this tag must be removed (Young et al. 2012 Biotechnol J. 7(5):620-34), and additional downstream processing may be required. We agree that more characterization is necessary to better assess the purity of our plant-produced products. To further characterize the rabies plantibodies, future studies will employ mass spectrometry and size exclusion chromatography.
Authors need to state whether the RABV isolate they used for challenge, TDRBV-QSMI2, has epitope incompatibility with at least one of the Mabs included in the cocktail. Based on the results, I assume that both epitopes are present on viral glycoprotein but this needs to be described.
Author response:
The challenge virus used in this experiment derived from animal isolated from Thailand. The whole genome sequence of this isolate will be published on GenBank from June 20, 2024 with accession number: BankIt2595250 SQ382-M7 ON808418. We followed strictly the WHO’s criteria for the challenge virus and the candidate Mabs used in the study. Both epitopes are replicated sequence on viral glycoprotein.
(World Health Organization.WHO consultation on a rabies monoclonal antibody cocktail for rabies post exposure treatment. Geneva: WHO, 23–24 May 2002. World Health Organization, Geneva, Switzerland, 2002. Available at: http://web.archive.org/web/20100525173651/http:// www.who.int/rabies/vaccines/en/mabs_final_report.pdf. Accessed 19 February 2014)
Abstract: Language needs a substantial improvement.
Author response:
We genuinely appreciate the reviewer’s comments. We have thoroughly proofread and corrected the abstract in the revised manuscript as per suggestion.
Lines 33-47: Language needs an improvement.
Author response:
We genuinely appreciate the reviewer’s comments. We have thoroughly proofread and corrected the language in Lines 34-47 of the revised manuscript.
Lines 66-68: I think authors mean that in the case if Mab binding epitope is not conserved in certain viruses, these viruses would not be neutralized, and as such, a cocktail of two or more non-competitive Mabs targeting non-overlapping epitopes would warrant a better degree of confidence. If I am wrong, I am in the camp of readers who would equally misunderstand what authors wanted to say here.
Author response:
We apologize for the confusion. The sentence was changed in Lines 67-69 of the revised manuscript as it was misleading.
Lines 68-69: these Mabs do not have specificity to antigenic sites I or II, their binding epitopes are located within these antigenic sites. Here and elsewhere, authors need to be clear and specific in the terminology they use.
Author response:
We apologize for the unclear terms. The sentence was changed in Lines 70-72 of the revised manuscript because it was ambiguous.
Line 87: What are “reduction determinants of clinical rabies”?
Author response:
Factors that reduce incident of clinical rabies.
Line 164: What are “virogenic properties”?
Author response:
We apologize for the confusion. We amended the wording into characterizing the onset of the neurological signs.
Line 173: I do not understand this: “Hamsters were recuperated after 4 h of post infection”.
Author response:
To constrain rabies virus infection after exposure correlates with a timely prophylaxis by the patient. In actual circumstance, patients do not receive PEP right away after the expose. There is gap of time to transport to visit the medical care center which is varied between hours. We imitated this “transportation time” for the patients into “4-hour recuperation postexposure” for the experimental animals before the treatment began.
Lines 174-175: what that “experimentally PEP” was? I see it in lines 247-252, but in the layout that authors chose (M+M first, Results and Discussion next), these technical details must be explained here, and only partially/essentially repeated in Results.
Author response: The technical details, such as the experimental PEP, were explained and added in Lines 179-185 of the revised manuscript in the Materials and Methods section and only essential details were partially repeated in the Results section.
Line 179: did hamsters really “survived the infection for 28 days”?
Author response:
Yes, they did survive 28 days post-infection. (All of the infected animals died within 7 to 13 days post-infection). However, to terminate the experiment, all surviving animals were euthanized at day-28 and were confirmed for rabies-negative with the brain testing.
Lines 190-193: Authors need to check their explanation of this equation. The statement in Line 190 “Animal equivalent dose (AED) = Human equivalent dose (HED) x Km value” cannot be in agreement with the statement in line 193 “Km for adult human = 37, Km for hamster = 5”. Usually, the most simple and conventional way is to divide the animal weight by human weight, and calculate the corresponding drug dose as in these examples: https://www.ncbi.nlm.nih.gov/pmc/articles/PMC4804402/.
Author response:
Dose translation between species is crucial. We used BSA (body surface area) normalization to convert the dose between different species. Km factor based on data from US-FDA Draft Guidelines to convert dose in mg/kg to dose in mg/m2.
(Center for drug evaluation and research, Center for biologics evaluation and research. (2002) Estimating the safe starting dose in clinical trials for therapeutics in adult healthyvolunteers, U.S. Food and Drug Administration, Rockville, Maryland, USA.)
Figure 1 and caption (lines 217-222): there are no labels in this figure, impossible to follow the caption. Also, in line 218 it says E550 although everywhere also it is about E559.
Author response:
We apologize for the typographical error. E550 was changed to E559 in Line 226 of the revised manuscript.
Table 1: no need in this table, it can be replaced by one short sentence within the paragraph above.
Author response:
Table1 removed per suggestion.
Lines 144-310: was there any statistical difference, either in survivorship or at least in the duration of incubation or clinical periods between the groups?
Author response:
We added statistical analyses on survival curves in comparison among the treatment groups per suggestion. (Line 202-206, 302-304, 312-314)
Figures 2, 3, 4: MS Excel used for depicting the survival curves is certainly inferior to other software (e.g. Graphpad Prism) in graphing survival curves. In these figures, it is hard to understand the gradual decreases of “Animal survival percentage” along several days observed in some groups. Also, as I have a hard time to imagine a survival of 120%, I do not understand why not to limit the axis to 100%. I recommend to re-do all figures as Kaplan-Meier plots and to compare different Kaplan–Meier curves by the log rank test, and the Cox proportional hazards test to show significant difference between treatment variants.
Author response:
Survival curves were amended using Prism GraphPad and with Log-rank test (Mantel-Cox) per suggestion.
Lines 312-331: language needs an improvement.
Author response:
We genuinely appreciate the reviewer’s comments. We have thoroughly proofread and corrected the language in Lines 316-334 of the revised manuscript.
Reviewer 2 Report
The research article entitled “Efficiency comparative approach of plant-produced monoclonal antibodies against rabies virus infection” by Lumlertdacha et al., describes in great detail the production, purification and whole preparation of two plant-produced monoclonal antibodies [MoAb] (E559 and 62-71-3) with an estimated broad neutralizing activity based on their binding activity towards two major antigenic sites within the receptor binding rabies virus (RABV) glycoprotein. Authors tested these two monoclonal antibodies combined and alone, in vitro and in vivo. In vivo, authors used them alone and, in a cocktail, together with rabies vaccine to test efficacy against a particularly high RABV challenge dose of a dog derived laboratory highly passaged street virus. Authors reported high antibody yields in the plant platform with a superior efficacy and potency alone and with rabies vaccine when compared in same combinations with human and horse derived immunoglobulins. Authors assembled a compelling and complete discussion incorporating an interesting exercise on production costs of currently available MoAb with those produced in plant-based platforms thereby supporting lower production costs that may support broader applicability perhaps in less resourceful countries.
The research is timing to contribute on the global goal on rabies elimination in humans transmitted by dogs by the 2030. Significant production-cost reductions make this platform a feasible choice to take this type of biologic to most vulnerable populations and in tur would help reduce remarkable global health disparities.
I just have two small comments or questions.
1) Notably, this monoclonal cocktail has significant production costs advantages over similar biologics and the ones produced in humans and horses. However, your results suggest that the most effective concentration dose for this plant-based cocktail seem to be double the concentration (40 IU/Kg), needed for the HRIG. Can you please briefly explain how this higher concentration needed would affect the final delivery price compared with that of HRIG?
2) Challenge virus. The challenge dose required to kill all controls was extremely high 43X MICLD50 of TDRBV-QSMI2? Can you express that dose as a back titer to have a sense how much RABV was actually inoculated to the hamsters?
3) Can please explain how the high passage number could affect the virulence of your challenge RABV (TDRBV-QSMI2)? In your methods you explain that the high passage number meat to increase virogenic (line 165) properties of the challenge virus. However, it is not clear what exactly authors meant with that term?
4) What virogenic properties did you tested and how you did it?
5) Did you test for increase of virulence, how you did it?
6) Have you considered that the high number of passages could attenuate your challenge virus and that is why you use such a high challenge dose?
7) Would you please elaborate on how the virogenic properties could affect your efficacy results?
8) Please elaborate on these questions within your discussion.

minor editions please see attached file
Author Response
Reviewer 2
The research article entitled “Efficiency comparative approach of plant-produced monoclonal antibodies against rabies virus infection” by Lumlertdacha et al., describes in great detail the production, purification and whole preparation of two plant-produced monoclonal antibodies [MoAb] (E559 and 62-71-3) with an estimated broad neutralizing activity based on their binding activity towards two major antigenic sites within the receptor binding rabies virus (RABV) glycoprotein. Authors tested these two monoclonal antibodies combined and alone, in vitro and in vivo. In vivo, authors used them alone and, in a cocktail, together with rabies vaccine to test efficacy against a particularly high RABV challenge dose of a dog derived laboratory highly passaged street virus. Authors reported high antibody yields in the plant platform with a superior efficacy and potency alone and with rabies vaccine when compared in same combinations with human and horse derived immunoglobulins. Authors assembled a compelling and complete discussion incorporating an interesting exercise on production costs of currently available MoAb with those produced in plant-based platforms thereby supporting lower production costs that may support broader applicability perhaps in less resourceful countries.
The research is timing to contribute on the global goal on rabies elimination in humans transmitted by dogs by the 2030. Significant production-cost reductions make this platform a feasible choice to take this type of biologic to most vulnerable populations and in tur would help reduce remarkable global health disparities.
I just have two small comments or questions.
1) Notably, this monoclonal cocktail has significant production costs advantages over similar biologics and the ones produced in humans and horses. However, your results suggest that the most effective concentration dose for this plant-based cocktail seem to be double the concentration (40 IU/Kg), needed for the HRIG. Can you please briefly explain how this higher concentration needed would affect the final delivery price compared with that of HRIG?
Author response:
We apologize for the confusion. Statistical analyses have been added together with this declaration (line 202-206, 302-304) expressing that the 20 iu/cocktail Mabs showed no sig different with the survival curves versus the 40 iu/cocktail Mabs. (Quantitative save)
Biologics manufacturing output of the plant-Mabs is from “in vitro” process which is potentially cheaper than the Human or Equine origin purified globulin in comparison volume per volume.
2) Challenge virus. The challenge dose required to kill all controls was extremely high 43X MICLD50 of TDRBV-QSMI2? Can you express that dose as a back titer to have a sense how much RABV was actually inoculated to the hamsters?
Author response:
We aware that susceptibility to RBV (rabies virus) infection of the laboratory-animal species are greatly varied. Not every rodent species is suitable for INTRA-MUSCULARLY challenge.
We also aware that the challenging dose depends on the site of the inoculation.
For intracranially challenge of Standard Rabies Virus in the mouse requires 12-50 times of MICLD50 per recommendation by WHO Expert Consultation on Rabies. (reference: Meslin, F. X, Kaplan, Martin M, Koprowski, Hilary & World Health Organization. (1996). Laboratory techniques in rabies, 4th ed. World Health Organization. https://apps.who.int/iris/handle/10665/38286 [page363])
However, we were experimenting an intra-muscularly infection on the hind-limb of hamster species within considerable dose (43 times of MICLD50). Viral dosage was critically titrated in the hamsters before the experiment initiated so that the onset of the incubation period is appropriated.
For example, these publication demonstrated an amount of 100 times MICLD50 of RBV for the challenging dose in hamster species (Tzu-Yuan Chao, et al.,In Vivo Efficacy of SYN023, an Anti-Rabies Monoclonal Antibody Cocktail, in Post-Exposure Prophylaxis Animal Models. Trop. Med. Infect. Dis. 2020, 5(1), 31; https://doi.org/10.3390/tropicalmed5010031) and a challenging dose of 103.1MICLD50 upto 103.4MICLD50 were applied in hamsters in similarly experiment (reference: Efficacy of rabies biologics against new lyssaviruses from Eurasia. Cathleen A. Hanlon ∗, Ivan V. Kuzmin, Jesse D. Blanton, William C. Weldon, Jamie S. Manangan, Charles E. Rupprecht. Virus Res 2005 Jul;111(1):44-54.)
3) Can please explain how the high passage number could affect the virulence of your challenge RABV (TDRBV-QSMI2)? In your methods you explain that the high passage number meat to increase virogenic (line 165) properties of the challenge virus. However, it is not clear what exactly authors meant with that term?
Author response:
We apologize for the confusion. We amended the wording “virogenic properties” into characterizing the onset of the neurological signs.
4) What virogenic properties did you tested and how you did it?
Author response:
We apologize for the confusion per explanation above (mention 3).
5) Did you test for increase of virulence, how you did it?
Author response:
We apologize for the confusion per explanation above (mention 3).
6) Have you considered that the high number of passages could attenuate your challenge virus and that is why you use such a high challenge dose?
Author response:
Challenging virus dosage was explained above (mention 2)
7) Would you please elaborate on how the virogenic properties could affect your efficacy results?
Author response:
We apologize for the confusion per explanation above (mention 3).
8) Please elaborate on these questions within your discussion.
Author response:
Thanks for the suggestion. We amended the wording “virogenic properties” into characterizing the onset of the neurological signs.